# IAAgent: Autonomous Inference Attacks Against ML Services With LLM-Based Agents

## Abstract

Inference attacks have been widely studied and offer a systematic risk assessment of ML services; however, their implementation and the attack parameters for optimal estimation are challenging for non-experts. The emergence of advanced large language models presents a promising yet largely unexplored opportunity to develop autonomous agents as inference attack experts, helping address this challenge. In this paper, we propose `IAAgent`, an autonomous agent capable of independently conducting inference attacks without human intervention. We evaluate it on 20 target services. The evaluation shows that our agent, using GPT-4o, achieves a 100.0% task completion rate and near-expert attack performance, with an average token cost of only $0.627 per run. The agent can also be powered by many other representative LLMs and can adaptively optimize its strategy under service constraints. We further perform trace analysis, demonstrating that design choices, such as a multi-agent framework and task-specific action spaces, effectively mitigate errors such as bad plans, inability to follow instructions, task context loss, and hallucinations. We anticipate that such agents could empower non-expert ML service providers, auditors, or regulators to systematically assess the risks of ML services without requiring deep domain expertise.

## 1 Introduction

The deployment of ML models in security-sensitive domains calls for a comprehensive understanding of potential risks during the inference phase. Inference attacks (IA), such as membership inference (Salem et al., 2019; Shokri et al., 2017; Song & Mittal, 2021) and model stealing (Carlini et al., 2020; Jagielski et al., 2020; Tramèr et al., 2016), are pivotal for assessing a model's robustness by highlighting vulnerabilities that could lead to sensitive information leakage. These vulnerabilities not only threaten privacy but also jeopardize the model owner's intellectual property (Cristofaro, 2020). Hence, ML service providers, third-party auditors, and even regulators are increasingly expected to assess the security and privacy risks of ML services. Despite their importance, conducting risk assessment via inference attacks remains challenging, as it requires detailed analysis, such as selecting the most appropriate shadow datasets.

This complexity presents significant hurdles for those without specialized expertise and demands considerable effort even from experienced practitioners. Recent progress in large language models (LLM) has introduced autonomous agents to automate complex tasks across various domains, such as web interactions (Zhou et al., 2023; Xie et al., 2024), data analysis (Cao et al., 2024; Lai et al., 2023), and ML experimentation (Huang et al., 2024). These agents have shown remarkable potential to reduce manual labor and improve efficiency (Lu et al., 2024; Liu et al., 2024a; Huang et al., 2024). However, our evaluation later demonstrates that current agent frameworks lack effectiveness in conducting risk assessment (see Section 4.1).

To fill this gap, we propose `IAAgent`, an autonomous agent tailored to automate the risk assessment of various inference attacks. Specifically, we focus on membership inference (Salem et al., 2019; Shokri et al., 2017; Song & Mittal, 2021; Nasr et al., 2019), model stealing (Carlini et al., 2020; Jagielski et al., 2020; Tramèr et al., 2016), data reconstruction (Fredrikson et al., 2015; Yin et al., 2020; Zhang et al., 2020), and attribute inference attacks (Melis et al., 2019; Song & Raghunathan, 2020; Song & Shmatikov, 2020). We present the details of each attack in Appendix C. The proposed agent acts as an independent expert in conducting risk assessments, dynamically adapt-

ing its behavior based on the basic information of the given target service and real-time execution feedback. In this way, it empowers non-experts to systematically assess the risks of ML services with minimal input and without requiring domain expertise. As shown in Figure 1, `IAAgent` comprises `ControllerAgent`, which manages attacks, and `AttackAgent`, which executes them. We further manually identify all critical steps in the assessment process and encapsulate each as a separate action with detailed guidelines to construct task-specific action spaces for the two agents. The environment is equipped with reusable resources, including Linux shells, starter scripts with implementations for different inference attacks, and datasets and models available for use.

We evaluate `IAAgent` on 20 target services. Our agent with GPT-4o achieves a 100.0% task completion rate, defined as the percentage of five runs in which all possible attacks are successfully executed. For comparison, the state-of-the-art MLAgentBench (Huang et al., 2024), originally designed for ML experimentation but adaptable for risk assessment in the same environment, achieves only a 26.3% completion rate. We further compare it with a human expert. They use ML-Doctor (Liu et al., 2022b), an assessment framework, to conduct inference attacks. We observe that `IAAgent` achieves near-expert performance. The average attack accuracy of `IAAgent` in conducting membership inference is only 1.0% lower than that of a human expert. Our agents are also cost- and time-efficient, with a token cost of $0.627 and 27.11 steps per run on average. In addition, we demonstrate that `IAAgent` can adapt a more optimized strategy for adaptive scenarios such as service constraints (e.g., query limitation). Closed-source models consistently outperform open-source models, but over time, open-source models, especially DeepSeek, have improved substantially. Through trace analysis, we identify four common types of errors in MLAgentBench: bad plans, inability to follow instructions, task context loss, and three types of hallucinations. We then illustrate how each major component of our design is intended to mitigate these errors.

**Contributions.** We summarize our contributions as follows:

- We propose the first autonomous agent, `IAAgent`, capable of automating risk assessment of inference attacks for the given ML services without human intervention.

- We evaluate the agent on 20 target services, demonstrating that our agent, powered by a robust LLM (e.g., GPT-4o), achieves near-expert performance at low cost ($0.627 per run).

- We present trace analysis to illustrate how each major component of our design contributes to the performance, and we summarize key lessons learned from the development process.

## 2 IAAGENT

### 2.1 MOTIVATIONS

Non-experts, such as ML service providers, third-party auditors, and regulators, are increasingly expected to assess the security and privacy risks of ML systems, yet they often lack the technical expertise needed to conduct risk assessments and interpret performance metrics effectively. This gap is further amplified by the complexity of determining optimal configurations on a case-by-case basis. Even with the attack implementation, candidate datasets $\{D_i\}_{i=1}^{|D|}$, where each $D_i$ may have $L_{D_i}$ annotated labels, and model architectures $\{M_i\}_{i=1}^{|M|}$ are available, the challenge still remains.

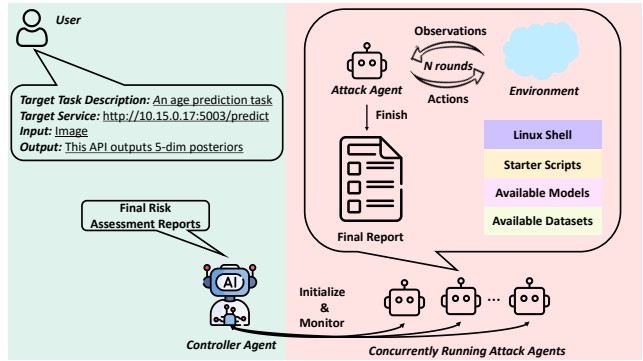

Figure 1: Workflow of `IAAgent`.

**Challenge #1: One must choose a dataset with a semantically similar label relevant to the same task, from at least $\sum_{i=1}^{|D|} L_{D_i}$ options.** This is non-trivial: it requires selecting a candidate

dataset whose data samples contain sufficient information to support the target task. Furthermore, one must ensure that the annotated labels within the selected dataset include one that is relevant to the target task, ideally with equivalent semantic meaning. In more complex cases, the inclusion of multiple labels per task results in a combinatorial explosion of possible options. In addition, the input modalities and output formats must be compatible; for instance, the selected label should have a similar number of classes to those used in the target model. Without this alignment, the shadow model may fail to replicate the decision boundaries of the target, significantly weakening the effectiveness of the attack.

**Challenge #2: Choosing an appropriate model involves balancing similarity and capability.** We take model stealing as an example. Powerful models can capture finer details in data, which help approximate the target model's outputs more precisely. However, using a model that differs significantly from the target model may reduce the fidelity of the imitation, thus weakening attack reliability.

**Challenge #3: Hyperparameter choices lead to a further combinatorial explosion.** Given a set of $n$ hyperparameters, each with $k_i$ possible values, the total number of hyperparameter configurations is $\prod_{i=1}^{n} k_i$, resulting in a combinatorial explosion in the search space. These include factors such as the dataset size and number of training epochs, each of which can have a substantial impact on the final outcome. For example, in membership inference, a smaller training dataset can make the shadow model more prone to overfitting, providing clearer signals for the attack to distinguish members from non-members. However, if the training dataset is too small, extreme overfitting could reduce the shadow model's ability to effectively mimic the target model's behavior.

**Challenge #4: The combinatorial explosion of dataset, model, and hyperparameter choices can overwhelm non-expert users.** While it is possible to enumerate all combinations across candidate datasets with their possible labels, model architectures, and hyperparameters, resulting in a search space of size $\sum_{i=1}^{|D|} L_{D_i} \times |M| \times \prod_{j=1}^{n} k_j$, selecting an optimal configuration from this space remains a substantial challenge, especially for those who may lack the experience or resources.

**Challenge #5: Effective result interpretation is essential for usability.** After completing the assessment, a report with intuitive explanations to interpret the attack results, along with corresponding defense suggestions to address these risks, is essential for those without specialized expertise to understand and manage the risks associated with target ML services.

## 2.2 DESIGN GOALS

The goal of `IAAgent` is to enable non-experts, such as ML service providers, third-party auditors, and regulators, to systematically assess the risks of a given ML service with minimal prior knowledge, while achieving near-expert-level performance. Rather than surpassing state-of-the-art attacks, we focus on facilitating assessments that help these non-experts identify vulnerabilities and make informed decisions about service deployment. We consider four goals for `IAAgent`:

Table 1: `ControllerAgent`'s action space.

| Action Name | Input | Observation | Purpose |
|---|---|---|---|
| Determine Attacks | A list of candidate attacks | A confirmation message | Confirm whether the given attack can be performed by the `AttackAgent` |
| Launch `AttackAgent` | Attacks to be launched | A message that the agent has been launched for each determined attack | Create an environment and launch an `AttackAgent` for each determined attack |
| Monitor Attacks | None | A status report of all attacks | Check the status of ongoing attacks |
| Final Answer | None | None | Shut down the agent and environment |

- **Minimal Knowledge Required**. The required knowledge includes black-box access to the target service, as well as basic information such as the task it performs and the general format of its inputs and outputs. This knowledge is commonly available to most potential users.

- **Near-Expert Performance**. `IAAgent` aims to complete risk assessment and achieve near-expert attack performance. This involves dynamically adapting its behavior based on both the basic information of each target service and real-time observations during execution.

- **Low Cost**. It also aims to accomplish the task with an acceptable API cost.

- **High Non-Expert Readability** Since our intended users may lack expertise in inference attacks and may not have the knowledge to conduct these attacks or interpret results solely based on metric

values, we provide them with an assessment report that clearly explains the attack process and the risks their service faces based on these results. We further provide alternative defense suggestions.

## 2.3 FRAMEWORK DESIGN

As illustrated in Figure 1, to conduct a risk assessment of their service, users need to provide basic information, including a brief description of the target tasks offered by the service (e.g., an age prediction task), the access (e.g., API endpoints), and the input and output formats. Optionally, the user can also provide a sensitive attribute (e.g., gender) that is not directly related to the original task of the target service to determine if there are unintentional leakages of this attribute. The assessment process proceeds without human intervention. The `ControllerAgent` receives basic information about the target service from users. It then determines which inference attacks it can perform and initializes `AttackAgent` concurrently to execute each attack. The `AttackAgent` conducts an inference attack following the instruction from `ControllerAgent`. After finishing the entire attack process, `AttackAgent` generates an easy-to-understand assessment report to interpret the attack results and explain the potential risk to the service. Both agents follow the same basic flow (see details in Appendix B.1), with a difference in the action space.

**ControllerAgent.** It is the entry point, and its main objective is to understand the current knowledge provided by the user, determine which attacks can be performed, and initialize `AttackAgent` for each one. The `ControllerAgent` continuously monitors the status of each running `AttackAgent` and exits the entire process once all `AttackAgents` have completed their tasks. Yang et al. (2024) demonstrate that important operations should be consolidated into as few actions as possible. Hence, we identify the critical steps and encapsulate each step as a separate action. Table 1 lists the actions available, and the details of each action are in Appendix B.2.

**AttackAgent.** The main objective of `AttackAgent` is to conduct an attack assigned by `ControllerAgent` and generate an easy-to-understand assessment report that summarizes the attack process and interprets the results. Similar to `ControllerAgent`, we identify all critical steps in the attack process, especially those related to challenging decisions as discussed in Section 2.1, and consolidate each into a separate action. Table 2 lists actions available for the `AttackAgent`. These actions include basic file system operations (e.g., reading files) and LLM calls. Moreover, as suggested by Cao et al. (2024), we provide domain knowledge as a guideline to help actions successfully progress toward their goal. More details of each action can be found in Appendix B.3.

**Environment.** Assessing the risk of the target service is a case-by-case task. Depending on the service, we may adjust shadow datasets, labels, model architectures, and hyperparameters for optimal assessment. While these choices vary, the inference attack workflow remains reusable. To streamline decision-making, we provide a reusable environment comprising four components: Linux shells, starter scripts, available datasets, and available models, to enable agents to accomplish their tasks more efficiently. More details of each component can be found in Appendix B.4.

## 3 EVALUATION

### 3.1 EXPERIMENTAL SETUP

**LLMs.** We evaluate `IAAgent` with closed-source representatives, including GPT-4o, GPT-4-Turbo, o3-mini, and Claude 3.5 Sonnet, as well as open-source representatives, including Mixtral-8×22B, Llama-3.1 (70B), and DeepSeek-V3.

**Target Service.** We train 20 different target models across five datasets and four model architectures. The dataset include CIFAR10 (Canadian Institute For Advanced Research, 2009), STL10 (Coates et al., 2011), CelebA (Liu et al., 2015), UTKFace (Zhang et al., 2017), and AFAD (Niu et al., 2016). The model architectures include Xception (Chollet, 2017), ResNet18 (He et al., 2016), ResNet50 (He et al., 2016), and a SimpleCNN consisting of three blocks of convolutional layers followed by a 2-layer fully connected network. We set up a web service based on (Flask, 2010) as the target service. This service loads the trained models from disk and provides API endpoints. For models trained on CIFAR10 and STL10, we expose a prediction endpoint that returns posterior probabilities, For models trained on the remaining three, in addition to the prediction end-

Table 2: Task-specific action spaces for `AttackAgent`. **C**$x$ denotes a solution to Challenge $x$.

| Action Name | Input | Observation | Purpose |
|---|---|---|---|
| List Files | A path of directory | A list of files and folders in the directory | List all files and folders in the given directory |
| Check Required Parameters | A script name | Descriptions of all required parameters | Extract all parameters required to be set when executing the given script |
| Choose Shadow Dataset | A file name, target task description, input format, output format, and target attribute | The name of the selected dataset with its path | Choose the most similar dataset as the shadow dataset (**C1**) |
| Choose Attribute | A file name, target task description, shadow dataset name, output format | The name of the selected attribute; If there are multiple attributes, return a string with attribute names separated by commas | Choose the most suitable attributes as the target labels for the shadow model (**C1**) |
| Choose Shadow Model Architecture | A file name, target service access, and the attack name | The name of the selected shadow architecture | Choose the most suitable model architecture (**C2**) |
| Set Parameters | A script name, a dataset name, a model name, an attack name, and the purpose of this script | Parameters with the exact values and concise reasons | Set the learning rate, batch size, number of epochs, and dataset size (**C3 and C4**) |
| Execute Script | A script name and a dictionary with parameter names and values | Any output from this execution | Execute the given script |
| Final Answer | None | None | Generate reports (**C5**) and shut down the agent |

point, we also expose an embedding endpoint that extracts representations from intermediate layers. See training details of the target models and deployment details of the target service in Appendix D.

**Environment.** The environment for the proposed agent includes the second half of the datasets and model architectures listed above. It also provides starter scripts for several attack methods: metric-based attacks (Salem et al., 2019) and neural-based attacks (Song & Mittal, 2021) for membership inference; standard model stealing and attribute inference attacks; and inversion alignment (Yang et al., 2019) for data reconstruction. Note that these attack methods only require black-box access.

**Evaluation Protocol.** The target services built on CIFAR10 and STL10 are expected to complete membership inference, model stealing, and data reconstruction attacks due to the absence of sensitive attributes (e.g., gender). Those built on AFAD, CelebA, and UTKFace are expected to complete all four types of attacks. We conduct five runs for each agent, allowing a maximum of 50 steps per run within a 5-hour runtime limitation. Our initial evaluation shows that increasing the number of steps or runtime does not improve effectiveness.

**Baseline.** To the best of our knowledge, there is currently no agent framework specifically designed to conduct risk assessment of ML services. Hence, we leverage the state-of-the-art agent from MLAgentBench (Huang et al., 2024), which is designed for ML experimentation, as a baseline. Its evaluation shows a better performance on 13 ML tasks than (LangChain, 2022) and (AutoGPT, 2023). To enable MLAgentBench to perform risk assessment, we provide it with the same environment, including the initial attack implementation, available datasets, and models.

**Compare With Human Experts.** The main goal of `IAAgent` is to accomplish the risk assessment as a human expert. Hence, we consider a human expert using ML-Doctor (Liu et al., 2022b), the state-of-the-art assessment framework, to conduct inference attacks. The expert, with knowledge of the training configuration of the target service, uses the disjoint set of the same dataset as the shadow dataset, along with the same shadow model architecture and training hyperparameters such as dataset size, batch size, and number of epochs. Note that this is the default setting in ML-Doctor.

**Evaluation Metrics.** We compare the baseline, `IAAgent`, and human from two perspectives: (1) *Task Completion Rate.* We define task success as whether the agent performs a complete risk assessment based on the information provided by the user. The task completion rate is the percentage over five runs for each target service. We assume that a human expert, when using the default tool, can achieve a 100.0% task completion rate. (2) *Performance Metric.* For membership inference, we use the highest attack accuracy. For model stealing, as our primary goal is to achieve the highest accuracy on the target task, so we consider the classification accuracy on the evaluation dataset. For attribute inference, we consider the accuracy of inferring the target attribute as the performance metric. For data reconstruction, we consider the average mean square error (MSE) between the reconstructed images and the images from the target training dataset. For the first three performance metrics, higher values are better; for the last performance metric, lower values are better.

**Cost Calculation.** We consider the time, steps, and token consumption by the agent to complete the risk assessment, with token consumption converted to actual API cost based on current pricing.

Table 3: Attack performance of our agents with GPT-4o and a human expert using ML-Doctor (Liu et al., 2022b) on 20 target services (Section 3.1). We run five rounds for each target and take the average values. "-" denotes *not applicable*.

| Target | | Membership Inference (↑) | | Model Stealing (↑) | | Data Reconstruction (↓) | | Attribute Inference (↑) | |
|---|---|---|---|---|---|---|---|---|---|
| Dataset | Model Arch. | Ours | Expert | Ours | Expert | Ours | Expert | Ours | Expert |
| CIFAR10 | Xception | 0.855 | 0.867 | 0.525 | 0.460 | 0.05218 | 0.05594 | - | - |
| | SimpleCNN | 0.761 | 0.762 | 0.585 | 0.557 | 0.05045 | 0.05046 | - | - |
| | ResNet18 | 0.855 | 0.859 | 0.543 | 0.492 | 0.05176 | 0.05198 | - | - |
| | ResNet50 | 0.850 | 0.855 | 0.545 | 0.485 | 0.04949 | 0.05010 | - | - |
| STL10 | Xception | 0.893 | 0.916 | 0.444 | 0.424 | 0.05332 | 0.04901 | - | - |
| | SimpleCNN | 0.762 | 0.750 | 0.455 | 0.465 | 0.05541 | 0.05009 | - | - |
| | ResNet18 | 0.888 | 0.902 | 0.469 | 0.439 | 0.05266 | 0.04809 | - | - |
| | ResNet50 | 0.880 | 0.900 | 0.446 | 0.428 | 0.05101 | 0.04625 | - | - |
| AFAD | Xception | 0.895 | 0.890 | 0.329 | 0.333 | 0.04791 | 0.04667 | 0.637 | 0.647 |
| | SimpleCNN | 0.872 | 0.879 | 0.325 | 0.328 | 0.04745 | 0.04655 | 0.672 | 0.710 |
| | ResNet18 | 0.918 | 0.945 | 0.339 | 0.341 | 0.04718 | 0.04731 | 0.754 | 0.772 |
| | ResNet50 | 0.914 | 0.925 | 0.348 | 0.335 | 0.04717 | 0.04578 | 0.693 | 0.717 |
| CelebA | Xception | 0.829 | 0.834 | 0.559 | 0.560 | 0.07200 | 0.07223 | 0.790 | 0.891 |
| | SimpleCNN | 0.735 | 0.735 | 0.518 | 0.353 | 0.06618 | 0.06579 | 0.878 | 0.884 |
| | ResNet18 | 0.815 | 0.869 | 0.524 | 0.507 | 0.06618 | 0.06766 | 0.884 | 0.894 |
| | ResNet50 | 0.828 | 0.831 | 0.558 | 0.536 | 0.06008 | 0.06314 | 0.885 | 0.888 |
| UTKFace | Xception | 0.722 | 0.725 | 0.733 | 0.708 | 0.04258 | 0.04283 | 0.569 | 0.573 |
| | SimpleCNN | 0.711 | 0.718 | 0.705 | 0.663 | 0.04137 | 0.04169 | 0.579 | 0.631 |
| | ResNet18 | 0.746 | 0.746 | 0.728 | 0.706 | 0.04143 | 0.04245 | 0.713 | 0.727 |
| | ResNet50 | 0.717 | 0.726 | 0.726 | 0.715 | 0.04349 | 0.04253 | 0.632 | 0.642 |
| **Average** | | 0.822 | 0.832 | 0.520 | 0.492 | 0.05197 | 0.05133 | 0.724 | 0.748 |

## 3.2 MAIN EVALUATION

**IAAgent achieves 100.0% task completion rate.** Our `IAAgent` achieves a far better task completion rate of 100.0% compared to the baseline, which only has 26.3% on average. See more details in Appendix E.1. We discuss common errors of the baselines and how we mitigate in Section 4.

**IAAgent achieves near-expert attack performance.** We present the performance of the proposed agents with GPT-4o. Due to the low task completion rate of the baseline, we decide not to compare against it. As illustrated in Table 3, our agents have near-expert performance compared with the human expert using ML-Doctor (Section 3.1). On average, our agents' maximum attack accuracy in conducting membership inference is only 1.0% lower than that of the human expert, and in conducting attribute inference attacks, the difference is only 2.4%. In model stealing, our agents even outperform the human expert by 2.8%. We attribute the good performance to our step-by-step guidelines encompassed in each task-specific action. For example, they instruct the agent to choose the most similar datasets in terms of target task, concept relevance, target label/attribute, and input/output formats. Even when the target labels used by the target service are not explicit, the guidelines can help the agent match the number of classes. Furthermore, in model stealing, the guidelines instruct the agent select more powerful model architectures and choose a larger number of training epochs and dataset size, leading to even better performance than the human expert. Note that the human expert follows the default settings in ML-Doctor (Liu et al., 2022b) without per-case optimization, as this would require substantial manual effort and make the cost difficult to measure.

**IAAgent only costs \$0.672 per run.** On average, `IAAgent` with GPT-4o spends about 147,971 input and 25,665 output tokens for each assessment. Converting with the current API prices (OpenAI, 2025), each run costs \$0.627. With its much lower task completion rate, the baseline's expected cost per run becomes \$0.873. See more details of token usage in Appendix E.2. We further present the distribution of the time spent and steps in Appendix E.3. `IAAgent` takes 27.11 steps and 17.39 minutes per run, while the baseline takes 32.67 steps and 21.05 minutes per run, even though there are many cases where it ends early before completion or reaching the maximum steps.

Table 4: Model stealing attack performance when the `IAAgent` adopts random selection and importance-based optimization strategies under a 3000-query limitation.

| | CIFAR10 | STL10 | AFAD | CelebA | UTKFace |
|---|---|---|---|---|---|
| Random | 0.522 | 0.422 | 0.312 | 0.348 | 0.646 |
| Optimization | 0.535 | 0.445 | 0.322 | 0.420 | 0.656 |

**IAAgent is extendable with specific actions for adaptive scenarios.** We consider the target service to have a 3,000 query limit and evaluate `IAAgent` in a model stealing experiment. We equip `IAAgent` with an importance-based optimization strategy (Wen et al., 2025). This study

shows that higher-importance data yields better attack performance. Therefore, we include an action that calculates the importance of each data point in the shadow dataset to optimally select query samples. When the `IAAgent` is aware that there is a query limitation on the target ML service, it autonomously selects an appropriate dataset based on the given information and optimally chooses the best data samples. As illustrated in Table 4, it indeed helps the `IAAgent` achieve better performance. Notably, on the CelebA dataset, with the same data size, the attack performance improves from 0.348 to 0.420. We further demonstrate that the proposed agent is also resistant to prompt injection attacks (see details in Appendix E.4).

**Closed-source models consistently outperform open-source ones, but over time, open-source models have improved substantially.** We evaluate our agents with representative LLMs, including closed-source models (Claude 3.5 Sonnet, GPT-4-Turbo) and open-source models (Mixtral-8×22B, Llama-3.1 70B). In general, the two closed-source models perform well, with Claude 3.5 Sonnet achieving a 100.0% task completion rate and near-expert attack performance. The details of the attack performance and efficiency comparison can be found in Appendix E.5. In contrast, two open-source models are much more prone to triggering errors. They randomly make multiple fabricated assumptions, using actions, e.g., `Change Directory`, that do not exist in action spaces (Table 2). Worse yet, these open-source models appear less effective at resolving errors. A predominant action usually emerges, repeatedly appearing in the trace instead of consulting memory or the environment. For Mixtral, the most frequent action accounts for 88.6% of occurrences in attack traces on average, and for LLaMA 3.1, it reaches 72.9%. This leads to recurring errors until the maximum step is reached. Fortunately, future LLMs continue to improve in instruction-following capabilities and reduce hallucination. We test two recent models, o3-mini and DeepSeek-V3; o3-mini succeeded in 20/20 cases, and DeepSeek-V3 in 18/20, narrowing down the performance gap.

## 4 TRACE ANALYSIS

### 4.1 ERRORS OCCURRED IN BASELINES

MLAgentBench has a single agent. The actions, such as `Write File` and `Understand File`, are generally applicable to ML tasks. Such general actions may be well-suited for applying the agent to different types of ML tasks. However, the generalizability might also cause the agent to lack control in decision-making (Xia et al., 2024) and ignore critical steps in the tasks. We analyze the traces of MLAgentBench and identify four types of common errors:

**Bad Plans:** Agents make bad plans, which can lead to severe consequences such as incomplete assessment and logical errors. For example, they may generate inappropriate actions, such as `Final Answer` to end the assessment without performing any of the attacks. The agent may also produce inappropriate action inputs. For example, the action `Edit Script` receives the edit instruction `set the dataset path to eval_dataset`, which uses the evaluation dataset as the shadow dataset, causing logical errors. They also schedule attacks poorly. For example, it initially inspects all starter scripts and collects extensive information, which can overwhelm its memory and cause context loss and hallucinations during execution. See details of each example in Appendix F.1.

**Inability to Follow Instruction:** Agents produce incorrect action inputs that violate usage guidelines in the instruction, causing them to become consistently stuck in an environment error. For example, the agent ignores the initial instruction `insert the parameter value as the default values` and explicitly passes parameters to the script for execution. However, the `Execute Script` action in MLAgentBench does not support explicitly passing the parameter, leading to a persistent environment error (see examples in Appendix F.2).

**Task Context Lost:** Agents may lose the task context, especially during the debugging process. For example, the agent gets stuck in a loop, repeatedly missing required parameter values, correcting them, and then missing them again(see examples in Appendix F.3).

**Hallucinations:** Agents make fabricated assumptions about the environment. Through our trace analysis, we observe three types of hallucination in MLAgentBench: **Type-I** refers to generating non-existent action names; **Type-II** refers to making fabricated assumptions about action inputs; **Type-III** refers to generating fabricated performance values. See more details of these hallucination and how do we mitigate them in Appendix G.

## 4.2 BENEFITS OF OUR DESIGN CHOICES

**Component-I: Multiple Agents As The Basic Framework.** In MLAgentBench, each response contains a `Research Plan and Status` entry used to trace what has been done. This entry is highly detailed and interpretable, making it useful for guiding the agent. However, the assessment tasks are not trivial and require many steps to complete. Furthermore, once the agent enters the debugging process, the entry becomes excessively long, causing it to forget the instructions and task context, and sometimes even ending the task without completing it, as shown above. Huang et al. (2024) also mention this entry fails to prevent situations where the agent plans to carry out overly complex edits and becomes stuck in debugging.

To mitigate such errors, we leverage the multi-agent framework. `ControllerAgent` determines which attacks to perform and launch `AttackAgent` for each attack. Each `AttackAgent` is responsible for only one attack and maintains its own memory independently. This framework prevents the agent from maintaining an overly long memory. It also ensures that

Table 5: The impact of different components in `IAAgent` on the task completion rate.

| MLAgentBench | + C-I | + C-II | + C-III | IAAgent |
|---|---|---|---|---|
| 26.3% | 78.0% | 64.9% | 50.5% | 100.0% |

each attack is executed independently, preventing one attack from blocking the execution of others if it gets stuck and reducing the risk of unexpected errors from using intermediate results across attacks. Meanwhile, the `ControllerAgent` submits the final answer only if it confirms that all `AttackAgents` have completed their attacks, ensuring that no attacks remain unperformed. As illustrated in Table 5, the task completion rate can be increased to 78.0%. Note that while the multi-agent framework improves task completion rate, attack performance remains poor due to random selection of datasets, models, and hyperparameters. For example, on a target service using the CelebA dataset and CNN model, the membership inference attack achieves 0.191, and the model stealing attack reaches 0.205.

**Component-II: Task-Specific Action Spaces.** In MLAgentBench, the agent does not recognize critical steps in the assessment. For example, choosing a shadow dataset is a critical step (see Section 2). A similar shadow dataset is critical to obtain high attack performance (Wen et al., 2025). Moreover, when conducting attribute inference, a mandatory requirement is that the shadow dataset must contain the target attribute intended for inference. However, we observe cases where the target service performs a facial attribute task. In these cases, the agent directly selects CIFAR10 as the shadow dataset to perform the attribute inference attack, even though CIFAR10 lacks attribute information. This causes the agent to become stuck in the debugging process and unable to resolve the issue until the maximum step limit is reached. The agent may also choose inappropriate actions while carrying out its plan. For example, they may execute `Final Answer` to finish the assessment without performing any attacks, even though it claims to be preparing to perform them in the action input (see the example in Appendix F.1). We speculate that it might be because the agent does not find an appropriate action in action spaces. These failure modes motivate us to develop a task-specific action space in which we manually identify all critical steps and encapsulate each as a separate action. In each action, we follow the design principle (Cao et al., 2024) to provide a step-by-step guideline to help the action successfully progress towards the goal. As illustrated in Table 5, the task completion rate increases to 64.9%. More importantly, this design improves the attack performance. For example, in model-stealing attacks targeting a race classification service, MLAgentBench achieves 0.531 accuracy, whereas our design achieves 0.723.

**Component-III: Response Format With Record Information.** The agent may hallucinate about the environment due to missing relevant information in its memory (see examples in Section 4.1). This inspired us to add the entry `Important Information` in the response format to remind the agent that it has acquired important information, especially parameters with their values required when executing scripts. Specifically, this entry records all information about the target (e.g., the target task), as well as all paths (e.g., dataset paths) and names (e.g., attribute names) that appeared in observations or in previous steps. With this design, we indeed observe a reduction in hallucinations, and the task completion rate increases to 50.5%.

**IAAgent.** As illustrated in Table 5, these components are all indispensable, achieving the current 100.0% task completion rate and strong attack performance.

## 5 LIMITATIONS AND FUTURE WORK

We take an initial step toward developing an autonomous agent capable of conducting inference attacks without human intervention. As this is an early-stage effort, our primary focus lies in the agent's design, including a multi-agent framework, task-specific actions, and instruction prompting, in order to establish a solid foundation. We summarize the lessons learned in Appendix H. We acknowledge limitations that can be improved in future work as follows.

*More Target Models.* We did not target additional models, especially LLMs, because our main focus is on agent design rather than incremental extensions. Moreover, inference-time attacks on LLMs are typically evaluated using traditional reference-based methods, which are considered strong but require extensive resources that we acknowledge we do not have (Hayes et al., 2025).

*More Attacks and Methods.* Following previous work (Cristofaro, 2020; Duddu et al., 2024; Liu et al., 2022b), we focus on four representative inference attacks. However, our agent is flexible to perform other attacks. As an initial investigation, we test adversarial attacks (Goodfellow et al., 2015), with results presented in Appendix I. This demonstrates the feasibility of extending `IAAgent` to additional attacks; future work may develop richer action guidelines to enable more advanced attack techniques. Our evaluation does not cover all methods for each inference attack. While feasible, such incremental extensions are left as future work, as they lie outside the core contribution.

*More Adaptive Evaluation.* In Section 3.2 and our initial exploration of adversarial attacks, we show that `IAAgent` can perform adaptive evaluations under service-level defenses, such as query rate limits, and model-level defenses, such as adversarial training. The agent optimizes its strategy either by learning under known constraints or iteratively adapting when unaware of defenses. Future work could extend `IAAgent` with additional actions to handle a broader range of defense mechanisms.

*More Human Evaluation.* All authors reviewed the assessment report and found it accurately describes the attack process, interprets results, and provides defense suggestions. Future work will involve non-expert AI practitioners for a more systematic evaluation.

## 6 RELATED WORK

**Applications of LLM-Based Agents.** Researchers and industry practitioners have recently begun exploring the capabilities of LLM-based agents to tackle complex tasks across various domains (Zhou et al., 2023; Cao et al., 2024; Lai et al., 2023; Huang et al., 2024; Zhang et al., 2024; Yang et al., 2024; Jimenez et al., 2024; Bouzenia et al., 2024; Carlini et al., 2025). (Huang et al., 2024) propose MLAgentBench that includes 13 ML tasks to evaluate the capability of autonomous agents to solve these tasks. Yang et al. (2024) introduce SWE-agent that provides a set of agent-computer interfaces to facilitate LLM-based agents to autonomously solve software engineering tasks. Carlini et al. (2025) propose a benchmark to exploit the agent's capability to break adversarial example defenses. These agents are designed with different goals, they vary in their frameworks, available environment, tools, action spaces, and prompt designs. These differences help them perform well in their own tasks but limit their generalizability to other settings. Importantly, they cannot be directly applied to risk assessment. The most related work is by Carlini et al. (2025), but their goal is to break adversarial example defenses, and the agent in that setting has white-box access to both the model and the defense implementation.

## 7 CONCLUSION

We propose an autonomous agent `IAAgent`. It empowers non-experts to conduct risk assessment on a given ML service at a level comparable to human experts. It only requires black-box access and basic information, which are typically easy to obtain. We evaluate it on 20 target services built across five datasets and four model architectures. The evaluation demonstrates that our agent with a robust LLM (e.g., GPT-4o), across all target services, achieves a 100.0% task completion rate and near-expert attack performance, with an average token cost of only $0.627 per run.

## ETHICS STATEMENT

During the evaluation, all datasets are research-oriented and publicly available, so there is no risk of users being de-anonymized. Therefore, our work is not considered human subjects research by our institutional review board. We further build the target services with exposed APIs to perform the evaluation, rather than directly targeting real-world ML services, to avoid causing any harm to them. In this paper, we aim to design an autonomous agent that assists the owners of ML services in understanding the potential security and privacy risks during the inference phase. To avoid being exploited by a malicious adversary, the proposed agent framework will have the request-access feature enabled, and we will manually review applicants' information.

## REPRODUCIBILITY STATEMENT

We are committed to sharing our artifacts to promote research and the development of effective, cost-efficient autonomous agents, particularly for automating labor-intensive tasks. They include the evaluation datasets and code for the proposed agent framework and the deployment of the target service.

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

## A  THE USE OF LARGE LANGUAGE MODELS

The authors used an LLM to check for typos and grammar and polished the wording.

## B  DETAILS OF AGENT DESIGN

### B.1  AGENT WORKING PIPELINE

Figure 2 presents an example of the basic flow of `AttackAgent`. It acts on memory $m_t$, which consists of an initial instruction $m_0$ and the plans and observations from the last three time steps, to produce the plan $p_t$. The plan includes an action $a_t$ from the action space in Table 2. It then executes the action in environment $e_{t-1}$ to produce an observation $o_t$ and an updated environment $e_t$. Finally, it updates the memory based on the plan and observation at the current time step $t$. The `ControllerAgent` follows the same basic flow, with a difference in the action space (see details in Table 1).

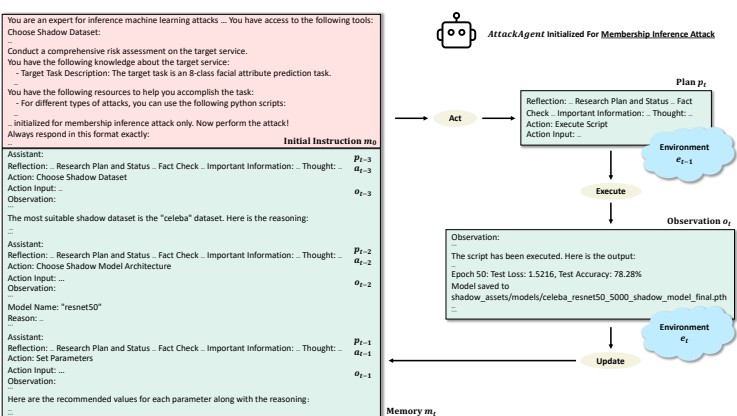

Figure 2: An example of the basic flow of `AttackAgent`.

### B.2  CONTROLLERAGENT'S ACTION SPACE

*Determine Attacks.* This action aims to determine which attacks to perform by the `AttackAgent`. It takes a list of attack names that `ControllerAgent` determines to perform based on the given information about the target service and returns the attack names that have been confirmed to be executable by `AttackAgent`.

*Launch `AttackAgent`.* It aims to create an environment and launch `AttackAgent` for each determined attack. After successfully launching all agents, it returns a success message.

*Monitor Attacks.* It aims to monitor the ongoing progress of each `AttackAgent`. The `AttackAgent` sends a completion message to the `ControllerAgent` once the entire attack process is finished. The observation contains status reports (e.g., `Completed`) from all `AttackAgents`.

*Final Answer.* The `ControllerAgent` uses this action to shut down the environment after completing the assessment.

### B.3  ATTACKAGENT'S ACTION SPACE

*Check Required Parameters.* This action aims to check all required parameters that need to be set when executing the given script. It takes a script name as input, reads the script, and calls an LLM to summarize the required parameters needed to execute the script. It returns concise descriptions of all parameters, including their types and purposes.

*Choose Shadow Dataset.* This action aims to select the most similar dataset to the shadow dataset. It takes a file name (e.g., `available_datasets.json`) that includes information about all available datasets in the environment, a description of the target task (e.g., `an age prediction task`), the input format (e.g., `image`) and output format (e.g., `5-dim posteriors`), and the target attribute (e.g., `gender`) as input. The target attribute can be any sensitive attribute that is not directly related to the original task. It first reads the file and calls an LLM to summarize relevant

information about each available dataset, such as the number of classes, common tasks that can be performed, and the dataset path. Then, it calls an LLM to select the shadow dataset using a detailed guideline to assist in the decision-making process. The guideline instructs the LLM to consider factors such as the shadow dataset having the same target task, sharing similar concept relevance, having the same target label and target attribute, and having the same input and output formats.

*Choose Attribute.* This action aims to select the appropriate attributes as the target label for the shadow model. It takes a file name that includes all attribute information for the selected shadow dataset (e.g., `available_datasets.json`), the description of the target task (e.g., `a facial attribute prediction task`), the selected shadow dataset name, and the output format of the target service (e.g., `8-dim posteriors`) as input. It first reads the file and calls an LLM to summarize all available attributes with their number of classes in this shadow dataset. Then, it calls an LLM to select the most appropriate attribute(s), guided by a detailed instruction to aid the decision-making process. This guideline provides empirical insights on selecting attributes, especially when none of the available datasets clearly match the target service's task description and the target label. In such cases, we attempt to match the output format, i.e., the number of classes of the target label. For example, we may select multiple attributes and combine them into a single label with the same number of classes as the target label. This action returns the exact name of the selected attribute(s) along with the number of classes. If multiple attributes are selected, it returns a string with attribute names separated by commas.

*Choose Shadow Model Architecture.* This action aims to select the appropriate shadow model architecture. Inputs include a file (e.g., `available_models.json`) with architecture details, target service access type (e.g., black-box), and attack name (e.g., `model stealing attack`). It first reads the file and calls an LLM to summarize relevant information about each available architecture. Then, it calls an LLM to select the most appropriate shadow model architecture, guided by a detailed instruction. This guideline contains empirical insights on maximizing the performance of the given attack. For example, if the primary goal of model stealing attacks is to maximize accuracy, a more powerful architecture is beneficial (Krishna et al., 2020). Hence, the guideline recommends starting with a more complex model architecture in model stealing, but not one that is overly complex, to avoid overfitting. This action returns the name of the selected model architecture and the reason for the choice.

*Set Parameters.* This action aims to set hyperparameters, including the learning rate, batch size, number of epochs, and dataset size, for training either the shadow model or the attack model. Before executing this action, the agent needs to execute the action *Check Required Parameters* to confirm whether these parameters need to be set. It takes a script name (e.g., `train_shadow_model.py`), a dataset name (e.g., `UTKFace`), a model name (e.g., `ResNet50`), an attack name (e.g., `membership inference attack`), and the purpose of the script (e.g., `training a shadow model`) as input. Then, it calls an LLM to set the values of these hyperparameters guided by detailed instruction, including empirical insights on maximizing the performance of the given attack. For instance, in shadow model training for membership inference attacks, a smaller dataset can increase overfitting risk, aiding the attacker's ability to differentiate members. Yet, if too small, excessive overfitting could reduce the shadow model's ability to effectively mimic the target model's behavior. Therefore, the guideline recommends starting with a small dataset size and gradually increasing it if needed. This action returns all parameters with the exact parameter values and concise reasons for choosing each parameter value.

*Execute Script.* This action aims to execute the given scripts with parameters. It takes a script name and a dictionary consisting of parameter names and corresponding values as input. Then, it explicitly passes these parameters to execute the given script and returns any output from this execution.

*Final Answer.* It shuts down the environment after completing the entire attack process. To ensure non-expert readability, it also generates an easy-to-understand report of the attack, including a description of the target service (e.g., task description and access point), the attack process, attack results with metric values (e.g., attack accuracy), explanations (e.g., whether the results indicate high risks), and defense suggestions.

### B.4 ENVIRONMENT

Assessing the risk of the target service is a case-by-case task. Depending on the service, we may adjust shadow datasets, labels, model architectures, and hyperparameters for optimal assessment. While these choices vary, the inference attack workflow remains reusable. To streamline decision-making, we provide a reusable environment comprising four components:

**Linux Shells.** `IAAgent` is built on a Linux environment and equipped with Shells to run Python scripts.

**Starter Scripts.** Attack implementations are reusable across target services, removing the need to reimplement them each time. We provide starter scripts covering all attacks, along with functions to access available datasets and models for a stable attack workflow. This enables the agent to focus on making challenging decisions, such as selecting suitable datasets, models, hyperparameters, etc., rather than generating code for different methods. Similarly, (Huang et al., 2024) provides starter code and data to build MLAgentBench for ML experimentation.

**Available Datasets.** Many inference attacks rely on an auxiliary dataset to either train a shadow model that mimics the behavior of the target model (e.g., membership inference and model stealing) or to directly train the attack model (e.g., attribute inference). We equip the environment with available datasets that can be used as auxiliary (shadow) datasets. We also provide descriptions of all available datasets in a JSON file. The description of each available dataset includes the number of classes, input data size, class names, dataset path, and common tasks it can perform, helping agents select the most suitable dataset for conducting attacks.

**Available Models.** We also provide a JSON file that includes a list of models available in the environment, determined by the maximum GPU resources in the environment. Only models that can run in the current environment are included in the list to avoid CUDA out-of-memory errors.

## C DETAILS OF RISK ASSESSMENT

Inference attacks allow adversaries to learn sensitive information about the training data as well as the functionality or parameters within the models. Following previous work (Cristofaro, 2020; Rigaki & García, 2024; Duddu et al., 2024; Liu et al., 2022b), we mainly focus on four representative inference attacks: membership inference, attribute inference, data reconstruction, and model stealing attacks.

**Membership Inference Attack.** It is a common type of privacy attack aimed at determining if a specific data sample is part of a training dataset (Liu et al., 2022a; Shokri et al., 2017; Salem et al., 2019; Choo et al., 2021; Li & Zhang, 2021; Ye et al., 2022). An adversary has access to a target service and attempts to determine whether a given data sample $(x, y)$ is included in its training dataset. Most attack methods require the adversary to train shadow models that mimic the target model's behavior on a shadow dataset. The adversary is also provided with additional information about the data distribution $\mathbb{D}$, which facilitates the creation of the shadow dataset. The quality of the shadow dataset and the performance of the shadow models are crucial to the overall success of the attack.

**Model Stealing Attack.** This attack aims to replicate a shadow model that mimics the functionality of the target model (Carlini et al., 2020; Jagielski et al., 2020; Tramèr et al., 2016). The workflow is that the adversary leverages data samples $\{\boldsymbol{x}_k\}_{k=1}^n$ from a specific distribution and queries the target service to obtain outputs $\{\mathcal{P}(\boldsymbol{x}_k)\}_{k=1}^n$ as pseudo-labels. They then construct the training dataset $\mathcal{D} = \{\boldsymbol{x}_k, \mathcal{P}(\boldsymbol{x}_k)\}_{k=1}^n$ to train the shadow model that replicates the functionality of the target service. In this attack, the quality of the shadow dataset and the architecture of the shadow models are crucial to the overall success.

**Data Reconstruction Attack.** It aims to recover the data samples of the target training dataset (Fredrikson et al., 2015; Yang et al., 2019; Yin et al., 2020; Zhang et al., 2020). A representative workflow (Yang et al., 2019): The adversary first collects an auxiliary dataset $\{x_k\}_{k=1}^n$ from public sources based on background knowledge. This dataset is expected to share key features with the target service. The adversary then queries the target service with $\{x_k\}_{k=1}^n$ to obtain pre-

diction vectors $\{\hat{y}_k\}_{k=1}^n$, and uses the resulting pairs $\{(\hat{y}_k, x_k)\}_{k=1}^n$ to train an inversion model that learns to reconstruct inputs from outputs.

**Attribute Inference Attack.** It aims to predict sensitive attributes that are not directly related to the original task of the target task (Melis et al., 2019; Song & Raghunathan, 2020; Song & Shmatikov, 2020). For example, a target service for predicting ages may unintentionally predict race. The adversary collects a shadow dataset $\{x_k, y_k\}_{k=1}^n$ where $y_k$ is the sensitive attribute that they aim to infer. This dataset is used to train the attack model, a two-layer fully connected network, where the input is the embedding, and the model predicts the sensitive attribute. The adversary queries the target service with $\{x_k\}_{k=1}^n$ to obtain embeddings $\{\mathcal{E}(x_k)\}_{k=1}^n$ to construct the attack training dataset. The shadow dataset needs to include annotated target attributes, and its quality is also crucial to the success of the attack.

## D  DETAILS OF TARGET SERVICE

We consider five datasets including CIFAR10 (Canadian Institute For Advanced Research, 2009), STL10 (Coates et al., 2011), CelebA (Liu et al., 2015), UTKFace (Zhang et al., 2017), and AFAD (Niu et al., 2016). Each task corresponds to a downstream task as follows:

- **CIFAR10** is a benchmark dataset that contains 6,000 images for each of 10 classes. The target task is a 10-class image classification task that categorizes animals (e.g., cats and dogs) and transport tools (e.g., airplanes and ships).

- **STL10** is a 10-class image dataset that contains 1,300 images for each of 10 classes. The target task is a 10-class image classification that can classify airplanes, birds, cars, cats, deer, dogs, horses, monkeys, ships, and trucks.

- **CelebA** is a facial image dataset that contains more than 200,000 facial images, each annotated with 40 binary attributes. Following previous work (Zhong et al., 2022; Wen et al., 2025; Nguyen & Tran, 2021), we select the three most balanced attributes to create an 8-class facial attribute prediction task.

- **UTKFace** contains approximately 23,000 images, each annotated with three attributes: gender, race, and age. We consider 5-class race classification as the target task.

- **AFAD** contains more than 160,000 facial images, each annotated with age and gender attributes. We divide age values into five bins to create a 5-class age prediction task.

Each dataset is split into two halves: the first for training target models and the second for risk assessment. We randomly sample 3,000 samples from the first half of STL10 and 5,000 samples from the first half of each of the other five datasets to train separate target models for each dataset. We use cross-entropy as the loss function and Adam as the optimizer, with a learning rate of 1e-3 and a batch size of 64. Each target model is trained for 300 epochs. The deployment of target services with API endpoints is shown in Figure 3.

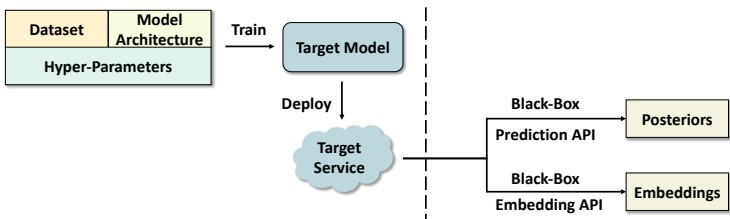

Figure 3: Deployment of target services with API endpoints.

# E    DETAILS OF EVALUATION RESULTS

## E.1    TASK COMPLETION RATE

We report the task completion rate over five runs where each agent successfully completes the entire assessment in Figure 4. Our `IAAgent` achieves a far better task completion rate of 100.0% compared to the agent in MLAgentBench, which only has 26.3% on average.

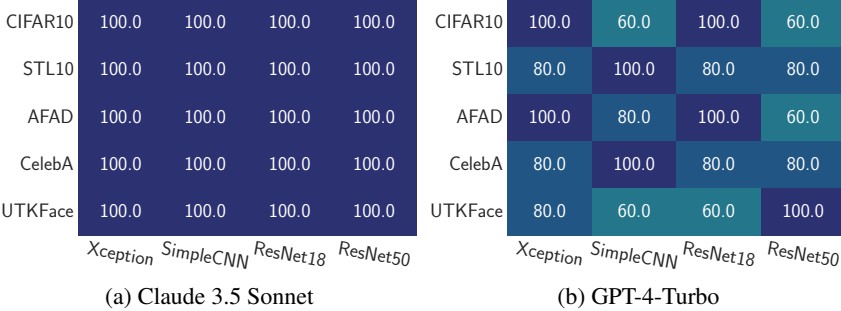

(a) Claude 3.5 Sonnet                                        (b) GPT-4-Turbo

Figure 4: Task completion rate of `IAAgent` with closed-source models, shown as the percentage over 5 runs in which the agent completes the entire risk assessment.

## E.2    DISTRIBUTION OF TOKEN USAGE

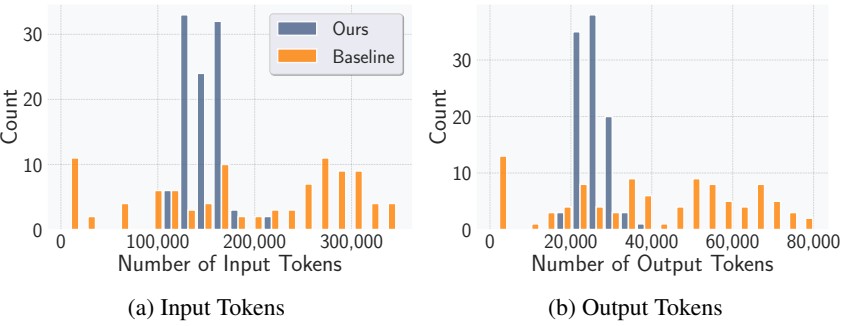

(a) Input Tokens                                              (b) Output Tokens

Figure 5: The distribution of token usage of the `IAAgent` and baseline with GPT-4o for each run.

We present the distribution of numbers of input and output tokens consumed by `IAAgent` in Figure 5. Our `IAAgent` demonstrates stable input and output token usage within a concentrated range, indicating that `IAAgent` is better optimized for consistent performance. In contrast, the baseline shows a much wider spread in both input and output token consumption, reflecting unstable and unpredictable task performance.

## E.3    DISTRIBUTION OF TIMES AND STEPS

We show the distribution of the time spent and steps taken by our `IAAgent` and the baseline for each run in Figure 6. We observe that our agents complete the assessment within a concentrated, shorter range of time and with fewer, consistent steps, indicating high efficiency. A few runs take longer, as they select a larger dataset size to improve the attack performance of model stealing attacks. In contrast, we observe that many runs fall into the first and last bins in Figure 6b. These correspond to cases where inappropriate actions lead to incomplete assessments, and where triggering numerous errors causes the agent to get stuck in the debugging process until the maximum steps are reached. On average, `IAAgent` takes 27.11 steps and 17.39 minutes per run, while the baseline takes 32.67 steps and 21.05 minutes per run, even though there are many cases where it ends early before completing the assessment or reaching the maximum steps.

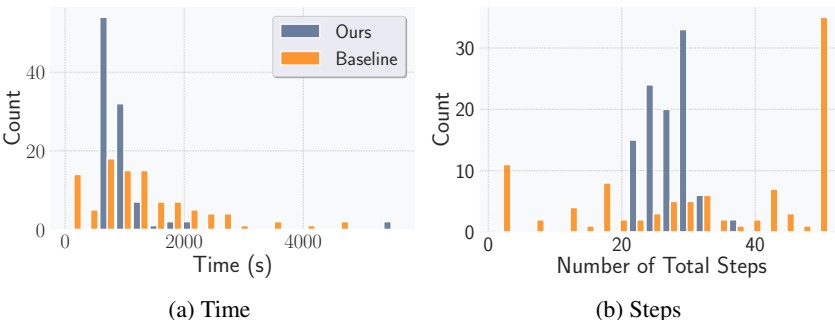

(a) Time            (b) Steps

Figure 6: The distribution of (a) time spent (b) steps taken by `IAAgent` and the baseline with GPT-4o for each run.

We further present the distribution of steps taken for each by `IAAgent` with GPT-4o in Figure 7. We observe that different attacks require varying numbers of steps to complete. On average, data reconstruction attacks require only 4.64 steps per run, the fewest steps, indicating a relatively straightforward attack. Membership inference attacks require more steps, 8.86 steps per run, indicating a higher complexity. In general, the steps taken for each attack are concentrated within a narrow range, demonstrating that `IAAgent` has considerable stability in achieving these attacks.

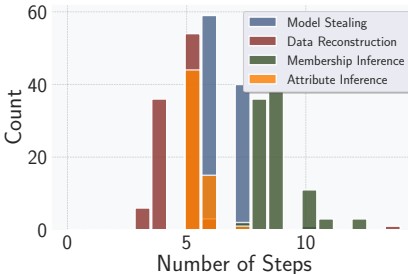

Figure 7: The distribution of steps taken by `IAAgent` with GPT-4o for each attack.

### E.4 ROBUSTNESS OF PROMPT INJECTION ATTACKS

**`IAAgent` is resistant to prompt injection.** Prompt injection attacks manipulate the agent's behavior through crafted inputs, corrupting its ability to perform the target task and causing it to execute attacker-desired actions. This can lead to harmful outcomes such as sensitive file leakage and unauthorized external requests (Liu et al., 2024b; Salem et al., 2023; Abdelnabi et al., 2023). We apply the standard prompt injection framework (Liu et al., 2024b) to assess `IAAgent`'s resistance to such misuse. Specifically, we assume a black-box adversary who can only manipulate inputs at the agent's entry point related to the target service. The adversary leverages native attacks, escape characters, context ignoring, fake completion, and combined attacks, and the injected task is spam detection. We observe that `IAAgent` successfully performs all attacks in all cases, i.e., 100.0% attack resistance. We attribute the high robustness to only the `ControllerAgent` receiving structured inputs (see Figure 2). Even with injected data, the `Important Information` entry only retained assessment-relevant details, and the `AttackAgent` launches and completes the assessment successfully.

### E.5 PERFORMANCE OF DIFFERENT LLMS

We compare the attack performance of `IAAgent` with different LLMs in Table 6. Here, for GPT-4-Turbo, we only include the attack performance in successful runs. We observe that, in successful runs, different LLMs achieve comparable attack performance across all target services.

Table 6: Attack performance of all four types of attacks launched by our agents with GPT-4o, GPT-4-Turbo (Turbo), and Claude 3.5 Sonnet (Claude) on target services built on five datasets and four architectures. We run five rounds for each setting and take the average values. "-" denotes *not applicable*.

| Dataset | Target Model Arch. | Membership Inference (↑) | | | Model Stealing (↑) | | | Data Reconstruction (↓) | | | Attribute Inference (↑) | | |
|---|---|---|---|---|---|---|---|---|---|---|---|---|---|
| | | GPT-4o | Turbo | Claude | GPT-4o | Turbo | Claude | GPT-4o | Turbo | Claude | GPT-4o | Turbo | Claude |
| CIFAR10 | Xception | 0.855 | 0.864 | 0.849 | 0.525 | 0.539 | 0.524 | 0.05218 | 0.05218 | 0.05218 | - | - | - |
| | SimpleCNN | 0.761 | 0.765 | 0.765 | 0.585 | 0.577 | 0.589 | 0.05046 | 0.05050 | 0.05051 | - | - | - |
| | ResNet18 | 0.855 | 0.863 | 0.869 | 0.543 | 0.540 | 0.546 | 0.05176 | 0.05176 | 0.05176 | - | - | - |
| | ResNet50 | 0.850 | 0.856 | 0.840 | 0.545 | 0.542 | 0.540 | 0.04949 | 0.04952 | 0.04952 | - | - | - |
| STL10 | Xception | 0.893 | 0.895 | 0.912 | 0.444 | 0.443 | 0.447 | 0.05332 | 0.05334 | 0.05332 | - | - | - |
| | SimpleCNN | 0.762 | 0.737 | 0.750 | 0.455 | 0.458 | 0.464 | 0.05541 | 0.05540 | 0.05540 | - | - | - |
| | ResNet18 | 0.888 | 0.889 | 0.889 | 0.469 | 0.461 | 0.462 | 0.05266 | 0.05266 | 0.05266 | - | - | - |
| | ResNet50 | 0.880 | 0.887 | 0.852 | 0.446 | 0.448 | 0.442 | 0.05101 | 0.05101 | 0.05101 | - | - | - |
| AFAD | Xception | 0.895 | 0.893 | 0.912 | 0.329 | 0.330 | 0.343 | 0.04791 | 0.04792 | 0.04792 | 0.637 | 0.647 | 0.639 |
| | SimpleCNN | 0.872 | 0.886 | 0.877 | 0.325 | 0.494 | 0.323 | 0.04745 | 0.04746 | 0.04747 | 0.672 | 0.595 | 0.663 |
| | ResNet18 | 0.918 | 0.940 | 0.938 | 0.339 | 0.340 | 0.345 | 0.04718 | 0.04717 | 0.04720 | 0.754 | 0.688 | 0.774 |
| | ResNet50 | 0.914 | 0.883 | 0.910 | 0.348 | 0.347 | 0.346 | 0.04717 | 0.04722 | 0.04722 | 0.693 | 0.585 | 0.700 |
| CelebA | Xception | 0.829 | 0.719 | 0.831 | 0.559 | 0.561 | 0.578 | 0.07200 | 0.07201 | 0.07201 | 0.790 | 0.800 | 0.888 |
| | SimpleCNN | 0.735 | 0.652 | 0.735 | 0.518 | 0.516 | 0.505 | 0.06618 | 0.06617 | 0.06617 | 0.878 | 0.727 | 0.728 |
| | ResNet18 | 0.815 | 0.829 | 0.866 | 0.524 | 0.523 | 0.521 | 0.06618 | 0.06618 | 0.06618 | 0.884 | 0.843 | 0.891 |
| | ResNet50 | 0.828 | 0.788 | 0.822 | 0.558 | 0.556 | 0.559 | 0.06008 | 0.06008 | 0.06008 | 0.885 | 0.888 | 0.702 |
| UTKFace | Xception | 0.722 | 0.722 | 0.732 | 0.733 | 0.725 | 0.727 | 0.04258 | 0.04262 | 0.04261 | 0.569 | 0.578 | 0.584 |
| | SimpleCNN | 0.711 | 0.713 | 0.712 | 0.705 | 0.706 | 0.709 | 0.04137 | 0.04138 | 0.04138 | 0.579 | 0.587 | 0.614 |
| | ResNet18 | 0.746 | 0.744 | 0.752 | 0.728 | 0.725 | 0.728 | 0.04143 | 0.04145 | 0.04145 | 0.713 | 0.714 | 0.693 |
| | ResNet50 | 0.717 | 0.723 | 0.723 | 0.726 | 0.726 | 0.727 | 0.04349 | 0.04354 | 0.04353 | 0.632 | 0.654 | 0.627 |
| Average | | 0.822 | 0.812 | 0.827 | 0.520 | 0.528 | 0.521 | 0.05197 | 0.05198 | 0.05198 | 0.724 | 0.692 | 0.709 |

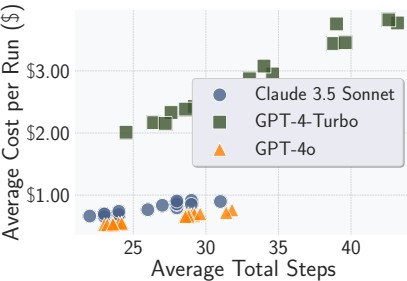

Figure 8: Comparison between `IAAgent` with different LLMs in terms of efficiency. Each point represents the average results of a target service over five runs.

We further compare the efficiency of `IAAgent` with different LLMs in Figure 8. Specifically, both GPT-4o and Claude 3.5 Sonnet are cost- and step-efficient and also demonstrate consistent performance. GPT-4-Turbo, while capable, incurs higher costs and requires more steps on average, making it less efficient for this task.

# F  MORE EXAMPLES OF ERRORS OCCURRED IN BASELINES

## F.1  BAD PLANS

As shown below, they may generate inappropriate actions, such as `Final Answer` to end the assessment without performing any of the attacks.

```
Action:  Final Answer
Action Input:  {
    "final_answer": "The feasible attacks given the access we have are:
        Membership Inference Attack, Model Stealing Attack, and Data
        Reconstruction Attack. Attribute Inference Attack is not feasible
        as it requires a black-box embedding API, which we do not have
        access to. We will proceed with preparing for these attacks by
        understanding the required parameters and setting them
        appropriately."}
```

As shown below, the action `Edit Script` receives the edit instruction `set the dataset path to eval_dataset`, which uses the evaluation dataset as the shadow dataset, causing logical errors.

```
Action:  Edit Script (AI)
Action Input:  {
    "script_name": "membership_inference/train_shadow_model.py",
    "edit_instruction": "Set default values for the required arguments:
        ... --dataset_path to 'eval_datasets', ...",
    "save_name": "membership_inference/train_shadow_model_with_defaults.
        py"}
```

## F.2  INABILITY TO FOLLOW INSTRUCTION

For example, as shown below, the agent ignores the initial instruction `insert the parameter value as the default values` and explicitly passes parameters to the script for execution. However, the `Execute Script` action in MLAgentBench does not support explicitly passing the `arguments` parameter as input, leading to a persistent environment error.

```
Action:  Execute Script
Action Input:  {
    "script_name": "data_reconstruction/inversion_attack.py",
    "arguments": {"--num_classes": 5,...}
}
Observation:
'''EnvError: The file data_reconstruction/inversion_attack.py",
    "arguments": {"--num_classes": 5,...} does not exist.'''
```

## F.3  TASK LOST

```
Research Plan and Status:
...
- Encountered an error due to missing required arguments during script
    execution.
- Edited the script to set default values for all required arguments,
    ensuring successful execution.
- Encountered a 'FileNotFoundError' due to missing shadow training
    dataset.
- Verified that the 'shadow_train_data/' directory does not exist in the
    current directory.
- Determined the correct path for the CelebA dataset as '
    supported_datasets/celeba_shadow.pt'.
- Successfully prepared the shadow dataset using the CelebA dataset path.
- Encountered an error due to missing required arguments during script
    execution.
- Inspected 'metric_based_attack.py' and confirmed it requires several
    command-line arguments.
- Edited 'metric_based_attack.py' to set default values for all required
    arguments.
- Encountered a 'FileNotFoundError' due to missing shadow training
    dataset.
...
```

## F.4  HALLUCINATION

As shown below, the agent sets the shadow dataset name as `default_shadow_dataset`, shadow dataset path as `path/to/shadow_dataset`, and the eval dataset directory `path/to/eval_dataset`.

```
Action:  Edit Script (AI)
Action Input:  {
```

```
    "script_name": "attribute_inference/attack.py",
    "edit_instruction": "Set default values for the required parameters:
        shadow_dataset_name='default_shadow_dataset', shadow_dataset_path
        ='path/to/shadow_dataset',..., eval_data_dir='path/to/
        eval_dataset'. Ensure these defaults are used if no command-line
        arguments are provided.",
    "save_name": "attribute_inference/attack_modified.py"}
```

## F.5  FAILURE MODES OF A SINGLE AGENT

In the following example, the agent completes membership inference, model stealing, and attribute inference, and then directly submits the final answer, stating that it did not perform the data reconstruction attack due to a lack of specific instructions or scripts for this attack. However, the `Research Plan and Status` records indicate that all four attacks can be performed.

```
Research Plan and Status:
...
2. Determine which attacks can be performed based on the available access
   .
    - We have access to both black-box prediction and embedding APIs, as
        well as white-box access to the model.
    - Status: All four types of attacks can be performed.
...
Fact Check:
- The successful execution of the 'attribute_inference/
    attack_final_corrected_v2.py' script and the final test accuracy of
    88.22% are directly confirmed by the observation.
Thought: ...
Action:  Final Answer
Action Input:  {
    "final_answer": "...Data Reconstruction Attack: Not performed due to
        the lack of specific instructions or scripts for this attack..."
}
```

## G  THREE TYPES OF HALLUCINATION

- **Type-I** refers to generating non-existent action names, such as `Change Directory` and `Review Code`, and attempting to have the agent execute them. It only appears in some specific LLMs, such as Mixtral and Llama-3.1 (see details in Section 3.2). By switching to stronger models (e.g., GPT-4o), we effectively reduce hallucinations.

- **Type-II** refers to making fabricated assumptions about action inputs. For example, the agent sets the shadow dataset as default_shadow_dataset, shadow dataset path as path/to/shadow_dataset, and the eval dataset directory path/to/eval_dataset (see examples in Appendix F.4). Although MLAgentBench can explore the environment to find the correct attack parameters and recover, it may get stuck in debugging, i.e., the inability to follow instructions and context loss, repeatedly reintroducing hallucinations. IAAgent decomposes tasks into sub-agents and leverages specific actions to guide the process, mitigating such issues.

- **Type-III** refers to generating fabricated performance values. It occurs when evaluation results are missing from the context, causing the agent to fabricate them. Explicitly recording this information effectively mitigates the issue.

## H  LESSONS LEARNED

We introduce an autonomous multi-agent framework IAAgent to conduct inference attacks. Our agent-based approach diverges fundamentally from simple scripting by incorporating a dynamic, iterative reasoning process that enables contextual understanding and adaptive problem-solving. This enables the synthesis of complex ideas, subtle decision-making, handling ambiguity or incomplete information, and effective error debugging and recovery, which are crucial for automated inference

attacks. Although `IAAgent` is designed for risk assessments, insights from its development can be applied to autonomous agents for other tasks.

First, a multi-agent framework breaks complex tasks into subtasks, with each agent handling a specific task and sharing memory only when necessary. This prevents overload and minimizes errors from shared results, allowing for more specialized instructions.

Second, we identify key steps in inference attacks and consolidate them into compact actions (Yang et al., 2024). These task-specific action spaces provide critical benefits by minimizing errors from overlooked critical steps and inappropriate actions, ensuring that agents progress toward the final goal. Step-by-step guidelines (Cao et al., 2024) improve stability and performance.

Third, a refined response format with key progress information helps mitigate errors, especially hallucinations. Robust LLMs (e.g., GPT-4o and Claude 3.5 Sonnet) are critical, as open-source models (e.g., Mixtral-8×22B and Llama-3.1) are more prone to errors and less effective at resolving them.

Fourth, reusable resources like starter scripts, datasets, and model architectures, establish a stable workflow, minimizing repetitive edits and allowing agents to focus on decisions.

## I EXTENSION TO ADVERSARIAL ATTACKS

we perform an initial investigation on adversarial attacks, which use carefully designed perturbations to inputs to reduce the accuracy of target models Goodfellow et al. (2015); Kurakin et al. (2016); Madry et al. (2018). We equip `IAAgent` with an initial implementation capable of performing the projected gradient descent (PGD) attack (Madry et al., 2018). We also incorporate basic actions, such as understanding and editing the script, into `IAAgent`. This allows the agent to iteratively tune the attack parameters of the initial script. We con-

Table 7: The attack performance of the initial PGD attack and the final PGD attack after optimization by `IAAgent`. The metric is the accuracy of the target service.

|  | CIFAR10 | STL10 | AFAD | CelebA | UTKFace |
|---|---|---|---|---|---|
| Initial | 0.450 | 0.399 | 0.369 | 0.317 | 0.423 |
| Final | 0.071 | 0.092 | 0.369 | 0.064 | 0.099 |

duct an evaluation on target services with CNN models. More notably, these models are trained with the adversarial training technique (Madry et al., 2018), a technique that improves model robustness by incorporating adversarial examples during training. `IAAgent` has no knowledge that the target model was trained with adversarial training. As illustrated in Table 7, we observe that, except for AFAD, the attacks successfully reduce accuracy to $< 10\%$. The agent effectively applied some strategies, such as tuning PGD hyperparameters (e.g., step count) and adding momentum. However, the agent's attempts to modify the PGD attack into more advanced attacks, such as C&W attack (Carlini & Wagner, 2017), fail and introduce numerous bugs.

