# OpenReview forum: "IAAgent: Autonomous Inference Attacks Against ML Services With LLM-Based Agents"
_ICLR.cc/2026/Conference — ICLR 2026 Conference Withdrawn Submission_

### Official Review · Reviewer_N4AE · 2025-10-27

**Soundness:** 3
**Presentation:** 2
**Contribution:** 2
**Rating:** 6
**Confidence:** 3

**Summary:**

This paper proposes the IAAgent, a multi-agent system designed to automate the evaluation of inference attack risk. They achieve a 100.0% task completion rate with GPT-4o and near-expert attack performance, at an average token cost of only $0.627 per run, demonstrating that it's highly effective and cost-effective.

**Strengths:**

- Provided sufficient context for inference attack and risk evaluation, and comprehensively presented the challenges of automating this evaluation process.
- Great results on the performance and the low cost.
- Show a fair amount of details in the appendix that support the main text.

**Weaknesses:**

Section 2.2 mentions that two of the goals of this system are (1) Near-Expert Performance and (2) High Non-Expert Readability. However,
- for (1), this work recruited only one human expert, which is insufficient to claim "Near-Expert Performance" in line 295.
- for (2), I do not see anywhere in the paper that it evaluates this.

**Questions:**

My questions are related to the two weaknesses I pointed out above:
- Can you justify why one human expert is sufficient to make that claim?
- I do not see anywhere in the paper that evaluates high non-expert readability. If this indeed is not evaluated, could you remove this part in section 2.2?

---

### Official Review · Reviewer_UruD · 2025-10-30

**Soundness:** 1
**Presentation:** 1
**Contribution:** 1
**Rating:** 0
**Confidence:** 4

**Summary:**

This paper considers the problem of creating an agent to automatically red-team AI systems for security vulnerabilities. Specifically, this paper develops an agent to automatically stress-test ML models for membership inference and model stealing attacks. The paper focuses on image classifiers as the ML system type that is being stress-tested. The results show that the proposed approach, an application of GPT-4o, out-performs all (weak) baselines and is very similar to human performance. The paper also provides examples the failures of baseline agents and benefits of the ControllerAgent/AttackAgent decomposition.

**Strengths:**

* The paper is overall well-written.
* The paper is well-motivated (not necessarily implying that this is relevant to the ICLR community).

**Weaknesses:**

* The paper does not provide enough background on the applications: membership inference attacks, model stealing, etc.
* The paper seems like a straightforward application of GPT-4o in conducting various AI security tasks. It's unclear that this provides any value to the ICLR community.
* The performance metrics are not well defined in the paper (lines 258-268 are not detailed enough) and the authors make little to no effort to find reasonable baselines.
* There are no examples of attacks generated by the agent.
* The action space of the agent is not well defined.
* Overall the paper is very opaque. Examples of the agent space and successful inference attacks would be beneficial.
* The related work is extremely impoverished.

**Questions:**

* Why would this paper be of interest to the ICLR community?
* Can you provide more details on the proposed approach?

---

### Official Review · Reviewer_VPvW · 2025-11-01

**Soundness:** 3
**Presentation:** 3
**Contribution:** 3
**Rating:** 4
**Confidence:** 4

**Summary:**

This paper introduces IAAgent, an autonomous agent capable of independently conducting inference attacks without human intervention. Experiments on 20 target services show that IAAgent archives a 100% task completion rate and near-expert performance using GPT-4o. IAAgent can also be driven by many representative LLMs and can adaptively optimize its strategy. Further analysis regarding the design choices such as a multi-agent framework demonstrates their effectiveness.

**Strengths:**

- This paper focuses on the application of LLM-based autonomous agents in specific areas. This is a hot research topic and will be even more important in the future.
- This paper is well-organized with clear illustrations, and is presented in a concise and accessible manner.
- The experiments are promising, indicating that IAAgent achieves high task completion rate and near-expert performance.

**Weaknesses:**

- Limited novelty. LLM-based autonomous agent has already been successfully applied in various areas [b1]. So it is not surprising that this technique would be helpful for inference attacks. IAAgent merely adapts this existing framework to the specific scenario of ML service inference attack assessment, without introducing new interaction logic or optimization mechanisms tailored to inference attack evaluation.
- In Figure 1, the workflow of IAAgent is mainly realized through Attack Agent and Controller Agent. The former executes specific inference attacks, while the latter coordinates the whole process. I don’t think that this work division framework is a very significant innovation. Rather, it is a very straightforward and apparent approach.
- From Appendix C, it seems that the four kinds of inference attacks considered in this paper are very old. The most recent methods are published in 2022. Many methods date back to 5 years ago or even 10 years ago. Considering that many new inference attacks have been proposed in recent years (from 2023 to 2025), this paper fails to analyze or integrate these latest attack paradigms, leaving gaps in its coverage of current ML service vulnerability scenarios and ultimately weakening the persuasiveness of its claim to provide systematic risk assessment.

[b1] Wang, Lei, et al. "A survey on large language model based autonomous agents." Frontiers of Computer Science 18.6 (2024): 186345.

**Questions:**

- From line 211 to line 213, it seems that only CNN-based models (i.e., Xception, ResNet18, ResNet50, SimpleCNN) are considered for the target models when evaluating the inference attacks. Given that Transformer-based models (e.g., ViT) have become widely adopted in various machine learning services (especially for vision-related tasks, which align with the datasets used here like CelebA and UTKFace), why are such Transformer-based models not included in the target model set? This exclusion raises questions about whether IAAgent’s performance and adaptability can generalize to ML services built on contemporary non-CNN architectures.
- This paper claims that IAAgent can adapt to service constraints like query limitations. However, in Table 4, it seems that only a fixed 3000-query scenario is tested for model stealing. Real-world ML services often have dynamic constraints (e.g., time-varying query limits, temporary API throttling). Why are such dynamic constraint scenarios not evaluated, and how to confirm IAAgent’s adaptability in these more practical, variable environments?

---

### Official Review · Reviewer_B7jJ · 2025-11-06

**Soundness:** 2
**Presentation:** 3
**Contribution:** 2
**Rating:** 4
**Confidence:** 3

**Summary:**

The paper proposes IAAgent, an autonomous system that runs inference attacks on ML services with little human input by combining a controller and attack agent, each operating over a task-specific action space and a reusable environment with shells, starter scripts, datasets, and models; the system targets membership inference, model stealing, data reconstruction, and attribute inference. The authors report experiments on 20 hosted services across five datasets and four architectures, claiming a 100% task completion rate with GPT-4o, near-expert attack performance. They further argue that multi-agent design, task-specific actions, and a response format with key facts reduce common failure modes seen in MLAgentBench, and that simple importance-based selection helps under query limits.

**Strengths:**

- Its studied problem is interesting: enabling non-experts to run standard inference attacks through an automated workflow.
- The query-budget study shows the framework can host adaptive strategies.

**Weaknesses:**

- Reported averages lack uncertainty (no CIs or multi-seed variance) and omit per-target paired analyses, so effect sizes and stability are unclear.
- The evaluation does not test live APIs, other modalities, or LLM targets, limiting external validity.
- Efficiency is framed as token and step counts, but there is no breakdown of wall-clock time by planning vs execution, nor sensitivity to model choice or prompt variants.

**Questions:**

- What is the measured gap to a tuned human expert per target, rather than to an untuned default workflow; does “near-expert” still hold?
- How sensitive are attack scores and costs to different decoding settings and prompt variants?
- What is the true end-to-end runtime and dollar cost broken down by planning, environment actions, and LLM calls, and how does this scale with deeper plans, longer horizons, and different backbones?
- Does the importance-based selection remain better than strong alternatives (e.g., query-synthesis or adaptive reranking) under the same query budget, and how often does it pick harmful or redundant samples?

---

### Note · Authors · 2025-11-24

I have read and agree with the venue's withdrawal policy on behalf of myself and my co-authors.